# Concurrent Infection of Skunk Adenovirus-1, *Listeria monocytogenes*, and a Regionally Specific Clade of Canine Distemper Virus in One Gray Fox (*Urocyon cinereoargenteus*) and Concurrent Listeriosis and Canine Distemper in a Second Gray Fox

**DOI:** 10.3390/pathogens9070591

**Published:** 2020-07-21

**Authors:** David B. Needle, Jacqueline L. Marr, Cooper J. Park, Cheryl P. Andam, Annabel G. Wise, Roger K. Maes, Rebecca P. Wilkes, Eman A. Anis, Inga F. Sidor, Dalen Agnew, Julie C. Ellis, Patrick Tate, Abigail Mathewson, Christopher Benton, Robert Gibson

**Affiliations:** 1New Hampshire Veterinary Diagnostic Laboratory, University of New Hampshire, Durham, NH 03824, USA; Inga.Sidor@unh.edu (I.F.S.); Robert.Gibson@UNH.edu (R.G.); 2Animal Health Diagnostic Center, Cornell University, Ithaca, NY 14850, USA; jm2699@cornell.edu; 3College of Life Sciences and Agriculture, University of New Hampshire, Durham, NH 03824, USA; cjp1043@wildcats.unh.edu (C.J.P.); Cheryl.Andam@unh.edu (C.P.A.); 4Veterinary Diagnostic Laboratory, Michigan State University, East Lansing, MI 48824, USA; wisea@msu.edu (A.G.W.); maes@msu.edu (R.K.M.); agnewd@msu.edu (D.A.); 5Animal Disease Diagnostic Laboratory, Purdue University, West Lafayette, IN 47907, USA; rwilkes@purdue.edu; 6Pennsylvania Animal Diagnostic Laboratory System, University of Pennsylvania, Philadelphia, PA 19104, USA; eanis@vet.upenn.edu; 7Northeast Wildlife Disease Cooperative, University of Pennsylvania, Philadelphia, PA 19104, USA; jellis04@vet.upenn.edu; 8New Hampshire Fish and Game, Concord, NH 03301, USA; Patrick.Tate@wildlife.nh.gov; 9Division of Public Health Services, New Hampshire Department of Health and Human Services, Concord, NH 03301, USA; Abigail.Mathewson@dhhs.nh.gov (A.M.); Christopher.Benton@dhhs.nh.gov (C.B.)

**Keywords:** Gray fox, canine distemper virus, adenovirus, skunk adenovirus, *Listeria*, listeriosis, zoonosis, wildlife

## Abstract

One free-ranging Gray fox (*Urocyon cinereoargenteus*) underwent autopsy following neurologic disease, with findings including morbilliviral inclusions and associated lesions in numerous tissues, adenoviral intranuclear inclusions in bronchial epithelial cells, and septic pleuropneumonia, hepatitis, splenitis, and meningoencephalitis. Molecular diagnostics on fresh lung identified a strain within a distinct clade of canine distemper that is currently unique to wildlife in New England, as well as the emerging multi-host viral pathogen skunk adenovirus-1. Bacterial culture of fresh liver resulted in a pure growth of *Listeria*
*monocytogenes*, with whole genome sequencing indicating that the isolate had a vast array of antimicrobial resistance and virulence-associated genes. One year later, a second fox was euthanized for inappropriate behavior in a residential area, and diagnostic workup revealed canine distemper and septic *L. monocytogenes*, with the former closely related to the distemper virus found in the previous fox and the latter divergent from the *L. monocytogenes* from the previous fox.

## 1. Introduction

Case 1: The carcass of an adult male Gray fox (Urocyon cinereoargenteus) from Alton, NH, was submitted for autopsy to the New Hampshire Veterinary Diagnostic Laboratory (NHVDL), having been euthanized because it was walking unafraid of humans, with limited use of one rear leg. The responding police officer easily approached and restrained the animal for euthanasia. 

Case 2: One year after the submission of case 1, the carcass of a Gray fox from Berlin, NH, was submitted to the NHVDL. The animal had been euthanized for inappropriate behavior, including a lack of fear of approaching humans, and sleeping in the open during daylight alongside a busy doorway of a house in a well-settled residential area. 

## 2. Materials and Methods

### 2.1. Pathology

The carcasses were subjected to a complete routine postmortem examination, with tissue samples both fixed in 10% neutral buffered formalin for histopathology and frozen fresh at −20 °F. Formalin-fixed tissues were routinely trimmed, processed, embedded, sectioned, and stained with hematoxylin and eosin. Histopathology was reviewed by board-certified veterinary anatomic pathologists. 

### 2.2. Rabies Virus Testing

Case 1 was not subjected to rabies testing, as no human or domesticated animal contact occurred and thus it did not satisfy the case definition for testing according to state regulations. Due to being located in a United States Department of Agriculture (USDA) wildlife services’ rabies surveillance zone, case 2 was subjected to rabies testing, via a direct rapid immunohistochemical test (dRIT) performed on a section of brainstem, as previously described [1].

### 2.3. Canine Distemper Virus Characterization

Frozen lung tissue from both animals was submitted for canine distemper virus (CDV) polymerase chain reaction (PCR). A portion of the M gene/M-F intergenic region was amplified and sequenced, as previously described [2]. Sequences from this region were submitted to GenBank and the submission also included sequences from this region for isolates previously described [3]. Phylogenetic analysis was performed by aligning these sequences with similar sequences obtained from GenBank using MAFFT Alignment version 7.450 [4]. An unrooted neighbor-joining tree was generated using the Geneious Tree Builder (Geneious Prime 2019, www.geneious.com/academic/) with Tamura-Nei nucleotide substitution model and 1000 bootstrap replicates.

### 2.4. Specific Canine Adenovirus 2 and Generic Adenovirus PCRs

Lung tissue from animal 1 was submitted for specific canine adenovirus 2 (CAV-2) PCR. An optimized real-time PCR assay was performed using primers specific for CAV-2 (forward primer, 5′-CTGACACTGCAATGCCTATATATATTTCCA-3′ and reverse primer, 5′-GACATAGAAACGCAGGACCCAGA-3′) and Taqman probe (FAM-5′-TGGCCATGCTAGCCACCCTCAGCCT-3′BHQ-1) [5]. The thermal cycling profile consisted of 45 cycles of denaturation at 95 °C for 10 s, annealing at 55 °C for 30 s, and extension at 72 °C for 10 s. 

A negative result for the specific CAV-2 PCR prompted the use of a generic adenovirus PCR assay, performed with slight modifications as described previously [6,7,8]. Tissue from animal 2 was not submitted for adenovirus PCR, as there were no lesions or intranuclear inclusions indicative of adenovirus infection in this animal. 

### 2.5. Bacterial Culture, Speciation, and Whole Genome Sequencing

Samples of fresh liver from each animal were cultured aerobically on Tryptic Soy with 5% sheep blood agar and MacConkey agar. The blood agar was incubated at 37 °C with 5% CO2, and the MacConkey was incubated at 37 °C without CO2. Bacterial isolates were identified using matrix-assisted laser desorption/ionization – time of flight (MALDI-TOF) mass spectroscopy (MALDI Biotyper^®^; Bruker Scientific, Billerica, MA, USA) as previously described [9,10]. Following MALDI-TOF biotyping, a subculture of the bacteria was sent to the NH Public Health Laboratory where DNA extraction was performed using the DNeasy blood and tissue kit (Qiagen, Hilden, Germany). Whole genome sequencing was performed on Illumina MiSeq (San Diego, CA, USA) following the PulseNet USA standard operating procedures (https://www.cdc.gov/pulsenet/pathogens/wgs.html) using the Nextera XT DNA library preparation and the 2 × 250 base pair sequencing chemistry. 

### 2.6. Genomic and Phylogenetic Analyses of L. monocytogenes

The *Listeria monocytogenes* dataset from a large study analyzing core genomes of 258 isolates, including those from 39 disease outbreaks, was used as a representative global population to compare to our genomes [11]. Of the genomes present in the dataset, 112 required assembly after retrieval. These genomes, as well as our own, were assembled using SPAdes v3.13.1 with default parameters [12]. All genomes were annotated using Prokka v1.14 with default parameters [13]. Roary v3.11 was used to cluster orthologous genes and identify the core genes shared by the genomes in the global dataset and the genomes of the two local isolates [14]. Sequences from each orthologous gene family were aligned using MAFFT [4]. Finally, the concatenated alignment of all core genes was used to generate a phylogenetic tree using RAxML v8.2.11 [15], with a generalized time-reversible nucleotide substitution model [16], four gamma categories of rate heterogeneity, and 100 bootstrap replicates. The phylogenetic tree was visualized using the Interactive Tree of Life (iToL) [17].

### 2.7. In Silico Identification of Resistome and Virulome Profiles

ABRicate was used to determine the resistance and virulence profiles of the two *L. monocytogenes* genomes [18]. Resistance genes were predicted by comparing sequences to the Resfinder database [19] using basic local alignment search tool + (BLAST+) [20], while virulence genes were identified using the virulence factor database (VFDB database) [21]. Genes were characterized as present (when 95% or more of the gene’s nucleotide sequence was present in the target genome with at least 95% sequence similarity), putative (<95% sequence coverage in the genome with 95% sequence similarity), or absent (0% coverage with 95% sequence similarity).

## 3. Results

### 3.1. Pathology

Case 1 was in reduced nutritional status, with adequate skeletal muscle and scant visceral fat. Gross examination showed dark pink mottling of the left lung lobes, miliary 1–2 mm diameter, round, pale tan, friable foci in the hepatic and renal parenchyma, and flocculent, red-brown urine.

Histopathology of the lungs revealed severe, multifocal, fibrinonecrotizing, neutrophilic, and lymphohistiocytic bronchointerstitial pneumonia. There was also fibrinous and fibrous, neutrophilic, and histiocytic pleuritis. Bronchial and bronchiolar epithelial cells often contained multiple small (2–5 µm diameter), round, eosinophilic, intracytoplasmic inclusions, and segments of epithelial cells contained karyomegalic nuclei filled with glassy, basophilic inclusions with peripheralized chromatin (Figure 1). 

There was fibrinonecrotizing, neutrophilic, and lymphohistiocytic meningitis of brainstem and cerebrum, with extension of the inflammation into Virchow-Robin space accompanied by adjacent gliosis and neuronal degeneration with satellitosis. Rare morbilliform intracytoplasmic inclusions as in the bronchi were seen in neurons and glia, as well as tracheal, biliary, urinary bladder, esophageal gland, gastrointestinal, and bulbar conjunctival epithelium, and stromal cells of a peripheral lymph node.

There were random foci of fibrinonecrotizing, neutrophilic, and lymphohistiocytic hepatitis and splenitis, with many bile casts in the liver. There was focally severe, bilateral necrotizing, neutrophilic interstitial cortical nephritis, and moderate secondary lymphoid tissue depletion.

Case 2 was in appropriate nutritional status. All lung lobes had foci that were rubbery and firm and did not collapse, alternating with pale pink, easily collapsed tissue. On cut surface, the firm foci were consolidated, mottled pink, red and purple, and oozed translucent, pale yellow fluid. The liver was mottled tan and pink, with an accentuated lobular pattern.

Histopathology revealed mild, subacute, lymphohistiocytic bronchointerstitial pneumonia, with erosion and intracytoplasmic morbilliform inclusions. The colonic Peyer’s patches displayed lymphocytolysis, which was moderate and acute, with intracytoplasmic morbilliform inclusions. Similar inclusions were noted in the biliary epithelial cell cytoplasm in the liver. Randomly scattered throughout the liver were small to moderately large foci of subacute necrotizing hepatitis, characterized by moderate, random foci of hepatocellular loss and replacement by amorphous necrotic debris, and infiltration by neutrophils and macrophages. There was also moderate, chronic, lymphoplasmacytic periportal hepatitis.

### 3.2. Rabies Virus Testing

dRIT testing on the brainstem from case 2 was negative. 

### 3.3. CDV Characterization

CDV M gene/M-F sequence placed the virus in both animals in a recently described clade that as of the time of manuscript submission has been described only in free ranging mesocarnivores in New England [3] (Figure 2: case 1 = Gray fox _9242, single arrowhead; case 2 = Gray fox_9243, double arrowhead). 

### 3.4. Skunk Adenovirus-1 Identification

In the diagnostic workup, based both on the historical reports of canine adenovirus in foxes in North America, and the fact that the animal was a canid, PCR specific for canine adenovirus was performed [22]. This test was negative. Generic adenovirus PCR, targeting the viral DNA polymerase gene performed on fresh lung tissue, amplified a ~320 bp fragment. Bidirectional Sanger sequencing of the PCR product yielded a unique 275 bp sequence. BLAST analysis of this sequence showed a 100% similarity to the corresponding sequence of skunk adenovirus-1 (SkAdV-1) DNA polymerase gene (GenBank Accession no. KP238322) [23]. 

### 3.5. Bacterial Culture, Speciation, and Whole Genome Sequencing

Aerobic cultures of the livers from both foxes yielded pure growth beta hemolytic bacterial colonies that were consistent with *Listeria monocytogenes.* Colonies were speciated using the MALDI-TOF Bruker Biotyper research use only (RUO) library current at the time of diagnosis (version 7 and 8 for the first and second cases, respectively), with species diagnosis confirmed via whole genome sequencing (WGS) [9,11]. When placed in the context of the *L. monocytogenes* from a global dataset, the phylogenetic tree built from core genes shows that the two *L. monocytogenes* isolates from the two foxes are not closely related to each other nor with other isolates from the United States [11]. The closest relatives of the two isolates are those from Switzerland and France (Figure 3). 

In isolate 1801300025 from the first Gray fox, in silico screening of its genome sequence revealed the presence of 13 genes associated with resistance against nine classes of antibiotics, including one multidrug resistance gene (Figure 4). In contrast, the isolate from the second fox (1904260017) carries only two resistance genes, conferring resistance against fosfomycin and lincosamide. In terms of genes associated with virulence, we identified a total of 183 genes that were present in the first isolate, while only 33 were found in the second isolate.

## 4. Discussion

SkAdV-1 was first identified in a skunk in Ontario Canada and was also previously described in a hedgehog in New Hampshire, United States [7,24]. To our knowledge, the initial fox we described in this study is the first free-ranging animal in New Hampshire diagnosed with SkAdV-1, and the first canid in any country reported with this virus [24]. It is also the second animal in New Hampshire, following a year after we detected SkAdV-1 associated with an outbreak of variably fatal respiratory in captive herd of hedgehogs [7]. In addition to the initial SkAdV-1 skunk, the hedgehogs, and our fox, and two porcupines in New York State, United States, with respiratory disease were found to have SkAdV-1 isolated from upper respiratory secretions [25]. Possible explanations for our detection of this recently and rarely described virus in one captive and one wild animal in New Hampshire within a brief period of time are that the virus is circulating undetected in wild skunks in New Hampshire or that there exists a previously unrecognized primary wild host/reservoir. The host range of SkAdV-1 is still being determined, and the addition of the canid reported herein to the previously described mustelid, erinaceids, and rodents indicates a wide infectivity. Such multi-host infection by individual adenoviruses has been purported to be associated with mutations or accelerated evolution in the natural history of these viruses [26,27]. 

The fact that SkAdV-1 was identified using degenerate primers after a negative PCR result for canine adenovirus highlights the importance of generic or degenerate PCR primers and their utility when specific molecular diagnostic test results are contrary to histopathology or other data. Without such generic/degenerate primers, we would not have identified SkAdV-1 in this fox or the hedgehog that we previously described [7]. 

In contrast to SkAdV-1, CDV is a well-known pathogen of carnivores. This *Morbillivirus* has been documented in many mammalian orders, including rodents, ungulates, and primates [28]. Gray foxes are susceptible to infection and significant morbidity and mortality, with a typical clinical presentation of neurologic disease, including depression, confusion, and lack of fear of humans, as well as gastrointestinal and respiratory disease [29]. As CDV is immunosuppressive, secondary infections are common. In measles, there is evidence that infection decreases the existing antibody library in an infected person, with mechanisms including reduction of reduced memory B cells, and increased numbers of T regulatory cells [30,31]. Such mechanisms may be at play in the foxes we described, with co-infection strikingly demonstrated by multiple individual bronchial epithelial cells containing both intracytoplasmic CDV inclusions and intranuclear SkAdV-1 inclusions (Figure 1). 

Numerous CDV strains have been identified in North American wildlife [32]. The strain detected in this case belongs to a distinct clade that has been reported in 8 carnivores in the Northeast United States in 2016–2017 [3]. This clade is distinct both from CDV strains used in commercial vaccines, and from others noted in wildlife in the United States [3,32]. The two CDV isolates from the foxes we reported herein are most closely related to each other and three isolates from raccoons from Rhode Island, USA. While vaccination has decreased CDV isolates circulating in domesticated dogs, viral transmission from wildlife reservoirs to unvaccinated or inadequately immunized dogs or animals in zoological collections is a threat [33]. The genetic characteristics of this strain could lead to infections in animals that are adequately vaccinated with commercial vaccines, as reported with infections of the America-4 strain circulating in Tennessee in domestic dogs [32]. 

While CDV has a wide mammal host tropism, *L. monocytogenes*, a small, Gram-positive, facultative pathogenic, saprotroph bacillus, is both ubiquitous in soil and known for its potential to cause multiple clinical syndromes, including abortions, encephalitis, and septicemia in mammals, humans, and birds [34,35]. In veterinary diagnostics, listeriosis is most often encountered in ruminants, often causing fatal encephalitis and metritis with resulting abortions. It has been reported in a myriad of species on every continent but Antarctica [36]. Listeriosis in humans is most often food-borne, affecting infants, the elderly, and immunosuppressed individuals, causing sepsis, gastroenteritis, and meningoencephalitis, all of which can be fatal [37]. Listeriosis is a reportable disease in the state of New Hampshire [38]. It is possible that this fox developed septic listeriosis secondary to its immunocompromised condition caused by CDV and/or SkAdV-1. There are historical case reports of similar co-infections of *L. monocytogenes* and CDV in Gray fox in New England [39,40]. The *L. monocytogenes* isolates in the two foxes we reported are distantly related to each other (Figure 3) based on the core genome tree and have clearly distinct virulence and resistance profiles (Figure 4), which likely indicates that they have disparate origins. While we have diagnosed CDV and SkAdV-1 in other wild or exotic animals in the state of NH in close temporal relationship to the cases we reported herein [3,7], *L. monocytogenes* has not been cultured in any non-domesticated animal besides these two Gray foxes in NH. This, in combination with the historical description of this pathogen isolated from Gray foxes [39,40], suggests the potential for a unique role of the Gray fox in the ecology of *L. monocytogenes*. 

## 5. Conclusions

The findings in these two foxes highlight the need for routine wildlife disease surveillance to improve our understanding and detection of new, emerging, and re-emergent strains of pathogens. In areas with high contact between wildlife species, humans, and domesticated animals, it is additionally important to monitor wildlife species for zoonotic agents to protect public health. The finding of septic listeriosis in this case poses a potential threat to humans and livestock, the distinct clade of CDV may pose a threat to domesticated dogs, and the identification of SkAdV-1 in a third order of mammals suggests wide infectivity of this emerging pathogen. Further, this study also highlights the value of using whole genome sequencing to characterize pathogens isolated from animals. 

## 6. Disclaimers

The opinions expressed by authors contributing to this journal do not necessarily reflect the opinions of the Centers for Disease Control and Prevention or the institutions with which the authors are affiliated.

## Figures and Tables

**Figure 1 pathogens-09-00591-f001:**
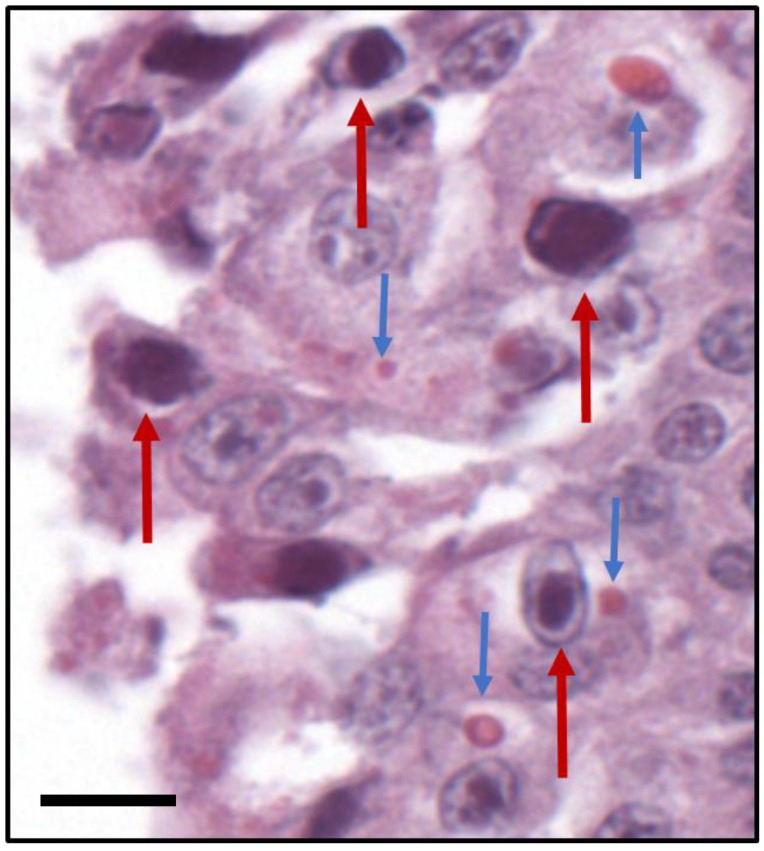
Concurrent canine distemper virus, skunk adenovirus-1, and *Listeria monocytogenes* in a Gray fox. Histopathology of bronchial epithelium. Bronchial epithelial cells containing glassine intranuclear inclusions with associated karyomegaly consistent with adenovirus (red arrows), and homogeneous eosinophilic 1–7 µm diameter round to oval intracytoplasmic inclusions consistent with CDV (blue arrows). Hematoxylin and eosin, 400×, scale bar 8 µm.

**Figure 2 pathogens-09-00591-f002:**
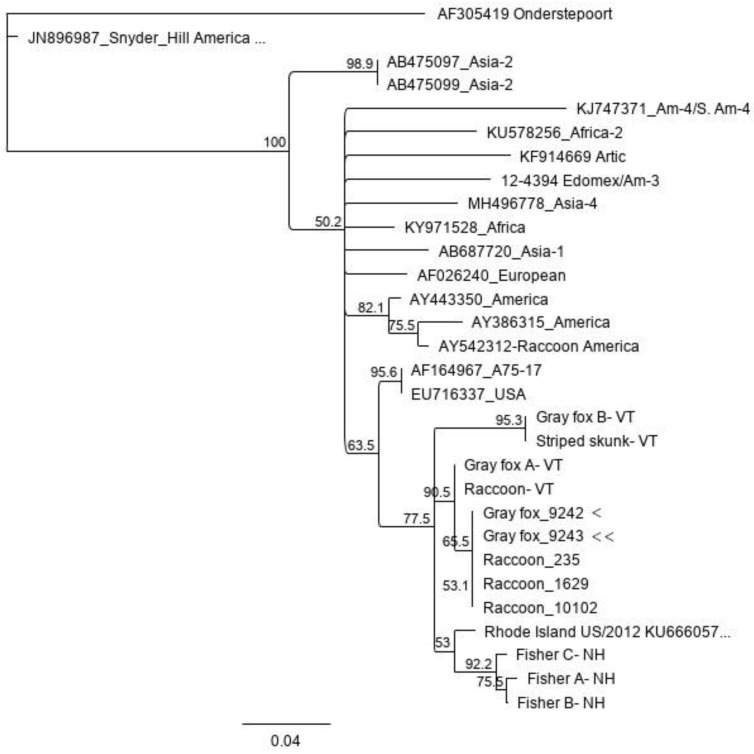
Canine Distemper Virus (CDV) M-F intergenic region phylogenetic tree. The two foxes described in this manuscript are Gray fox_9242 and Gray fox_9243 and are located with the recently identified distinct clade currently composed of isolates identified only in wildlife in New England. Geneious Tree Builder (Geneious Prime 2019, www.geneious.com/academic/) using the Tamura-Nei genetic distance model and an unrooted Neighbor-Joining tree build method with 1000 bootstrap replicates. Genbank accession numbers MN699664-MN699675 coincide with the two foxes describe in this report, Gray fox 9242 and Gray fox 9243, Raccoons 235, 1629, and 10102, and Fisher A- New Hampshire (NH), Fisher B-NH, Fisher C-NH, Gray fox A Vermont (VT), Gray fox B VT, Raccoon VT, and Striped skunk VT.

**Figure 3 pathogens-09-00591-f003:**
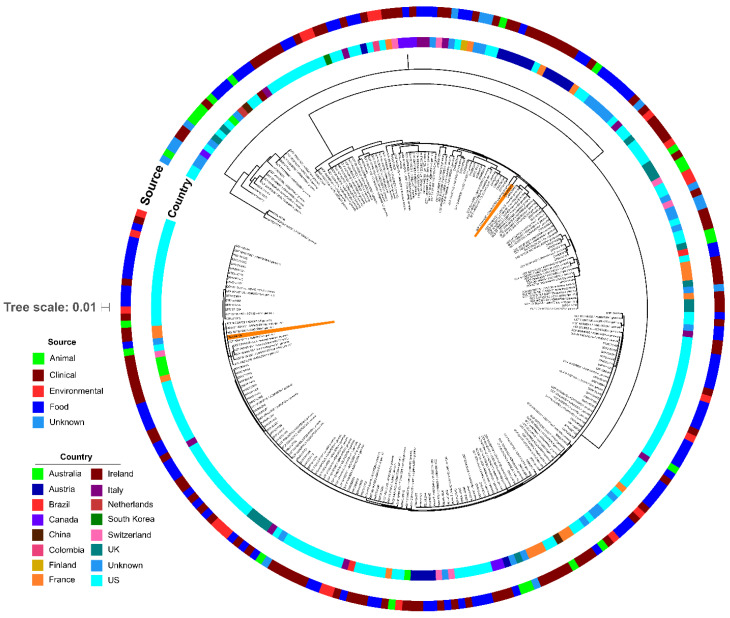
Phylogenetic tree of *Listeria monocytogenes* isolate core genomes. Concatenated alignment of all core genes was used to generate a phylogenetic tree using RAxML v8.2.11 with a generalized time-reversible (GTR) nucleotide substitution model, four gamma categories of rate heterogeneity, and 100 bootstrap replicates. The phylogenetic tree was visualized using the Interactive Tree of Life (iToL). The two bacterial isolates from the foxes are highlighted with orange and occupy nearly opposite positions on the tree.

**Figure 4 pathogens-09-00591-f004:**
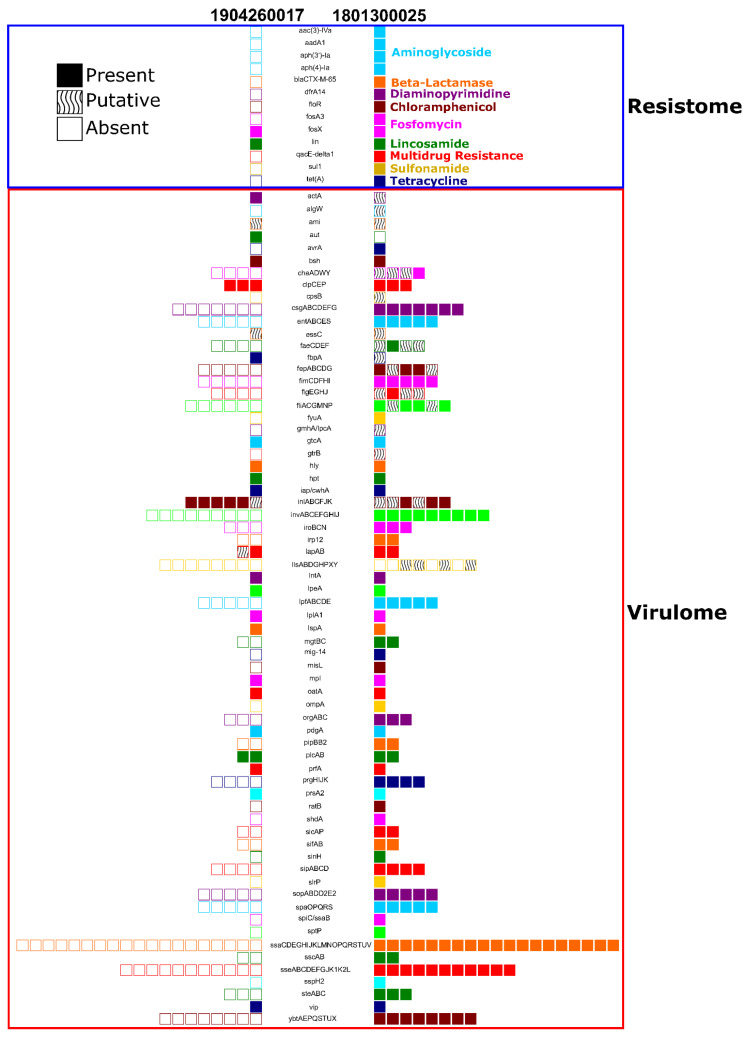
Resistome and virulome of *Listeria monocytogenes* isolates from one fox infected with *L. monocytogenes*, CDV, and SkAdV-1 (case 1 = 1801300025), and another infected with *L. monocytogenes* and CDV (case 2 = 1904260017). ABRicate was used to determine the resistance and virulence profiles by comparing resistance sequences in the Resfinder database using BLAST+ [8], and comparing virulence genes with the VFDB database. Genes were characterized as present (when 95% or more of the gene’s nucleotide sequence was present in the target genome with at least 95% sequence similarity), putative (<95% sequence coverage in the genome with 95% sequence similarity), or absent (0% coverage with 95% sequence similarity).

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
