# Peer review of "Concurrent Infection of Skunk Adenovirus-1, Listeria monocytogenes, and a Regionally Specific Clade of Canine Distemper Virus in One Gray Fox (Urocyon cinereoargenteus) and Concurrent Listeriosis and Canine Distemper in a Second Gray Fox"

_pathogens, 2020, doi:10.3390/pathogens9070591_

Round 1

Reviewer 1 Report

This manuscript written by Needle et al. presented a very well-documented case study under the topic “Concurrent infection of skunk adenovirus-1, Listeria monocytogenes and a regionally specific clade of canine distemper virus in one gray fox (Urocyon cinereoargenteus) and concurrent listeriosis and canine distemper in a second.”. Most of the analysis relevant to individual case is pretty well done, and the authors have certainly been very thorough. 

This is an interesting case study with useful data that may help many researchers and or countries to understand the importance of routine wildlife disease surveillance to improve the understanding and detection of new, emerging, and re-emergent strains of multiple pathogens.

One minor suggestion

The texts in the Figure 3 is not currently readable. Please check the file to make it clear.  

Author Response

Thank you for your time and review.

I am sorry the text is not legible in figure 3. I will re-insert the high resolution figure and also email a copy to the editorial team, in case they may be more successful in keeping the resolution. 

Reviewer 2 Report

  • line 38: put Introduction instead of “the subject”
  • the conclusions, although not mandatory, would be useful
  • "Biographical sketch" perhaps not necessary
  • the "Bibliography" chapter must be written according to the rules of the journal:

    https://www.mdpi.com/journal/pathogens/instructions

es. line 293, 303, 307, 310 etc. insert all authors;

line 348: missing vol. and article page

line 391: the vol. must be inserted after the name of the Journal

Author Response

Thank you for your time and review

  • line 38: put Introduction instead of “the subject”
    • I have made this change
  • the conclusions, although not mandatory, would be useful
    • I have made this change
  • "Biographical sketch" perhaps not necessary
    • I have removed this
  • the "Bibliography" chapter must be written according to the rules of the journal:

    https://www.mdpi.com/journal/pathogens/instructions

es. line 293, 303, 307, 310 etc. insert all authors;

line 348: missing vol. and article page

line 391: the vol. must be inserted after the name of the Journal

    • I have updated these. 

Reviewer 3 Report

The paper is well written and it is very interesting. 

The Authors should better describe in the introduction the historic of pathogens (CDV, SkAdV-1 and L. monocytogenes) circulation in the foxes's area of origin. 

In the "Discussion" section, the authors could speculate the epidemiologic role of foxes in the circulation and diffusion of the three patoghens considered.

Author Response

Thank you for your time and review.

  • The paper is well written and it is very interesting. 
    • Thank you!
  • The Authors should better describe in the introduction the historic of pathogens (CDV, SkAdV-1and L. monocytogenes) circulation in the foxes's area of origin. 
    • Since this is a case report, we chose to keep the format of introducing the two cases, and reserved the discussion of the pathogens for the discussion section.  
  • In the "Discussion" section, the authors could speculate the epidemiologic role of foxes in the circulation and diffusion of the three pathogens considered.
    • I will add a sentence regarding this, however until further studies are completed this would be just speculative, so I kept it brief and added it to lines 259-264.